# Coptisine Alleviates Imiquimod-Induced Psoriasis-like Skin Lesions and Anxiety-like Behavior in Mice

**DOI:** 10.3390/molecules27041412

**Published:** 2022-02-19

**Authors:** Ly Thi Huong Nguyen, Min-Jin Choi, Heung-Mook Shin, In-Jun Yang

**Affiliations:** Department of Physiology, College of Korean Medicine, Dongguk University, Gyeongju 38066, Korea; nguyenthihuongly_t58@hus.edu.vn (L.T.H.N.); zzz9924@dongguk.ac.kr (M.-J.C.)

**Keywords:** coptisine, psoriasis, anxiety, imiquimod, inflammation

## Abstract

Psoriasis is a common inflammatory skin disorder, which can be associated with psychological disorders, such as anxiety and depression. This study investigated the efficacy and the mechanism of action of a natural compound coptisine using imiquimod (IMQ)-induced psoriasis mice. Coptisine reduced the severity of psoriasis-like skin lesions, decreased epidermal hyperplasia and the levels of inflammatory cytokines TNF-α, IL-17, and IL-22. Furthermore, coptisine improved IMQ-induced anxiety in mice by increasing the number of entries and time in open arms in the elevated plus maze (EPM) test. Coptisine also lowered the levels of inflammatory cytokines TNF-α and IL-1β in the prefrontal cortex of psoriasis mice. HaCaT keratinocytes and BV2 microglial cells were used to investigate the effects of coptisine in vitro. In M5-treated HaCaT cells, coptisine decreased the production of IL-6, MIP-3α/CCL20, IP-10/CXCL10, and ICAM-1 and suppressed the NF-κB signaling pathway. In LPS-stimulated BV2 cells, coptisine reduced the secretion of TNF-α and IL-1β. These findings suggest that coptisine might be a potential candidate for psoriasis treatment by improving both disease severity and psychological comorbidities.

## 1. Introduction

Psoriasis is an immune-mediated inflammatory skin disorder that influences approximately 2% of the population, with a higher prevalence in Western countries [1,2]. The condition typically manifests as red, scaly, and thickened plaques on skin lesions caused by excess keratinocyte proliferation and immune cell infiltration [3]. The pathogenesis of psoriasis remains unclear but accumulating reports have suggested the crucial roles of keratinocytes and immune cells and their interactions in disease development [4,5]. Keratinocytes produce a variety of inflammatory factors, such as macrophage inflammatory protein 3α (MIP-3α or CCL20), interferon γ–induced protein-10 (IP-10 or CXCL10), and intercellular adhesion molecule-1 (ICAM-1) to recruit T cells to skin lesions [6,7,8]. The infiltration of T helper 1 (Th1) and Th17 cells resulted in high levels of inflammatory molecules, including tumor necrosis factor (TNF)-α, interferon (IFN)-γ, interleukin 17 (IL-17), and IL-22, which trigger the inflammatory response and excessive proliferation of keratinocytes [9].

Psoriasis is associated with psychiatric/psychological disorders, cardiovascular disease, and metabolic syndromes [10,11]. Anxiety and depression are the most common psychological comorbidities of psoriasis [11]. A multicenter observational study in 13 Western countries reported a 10.1% and 17.2% incidence of clinical depression and anxiety in psoriasis patients, respectively [12]. The elevated production of inflammatory cytokines observed in psoriasis, such as TNF-α and IFN-γ, might mediate the development of anxiety and depression [13,14]. Recently, there has been also a growing interest in drugs used by psoriasis patients and their effects on the psychiatric disorders that accompany psoriasis patients. While systemic corticosteroids have been associated with a high risk of depression and anxiety in psoriasis patients, biologics have been demonstrated to have the potential to reduce the risk of depression than conventional treatments such as methotrexate, cyclosporine, and systemic corticosteroids [15,16,17,18,19]. This increased the need to develop alternative medicines for psoriasis treatments to ameliorate the inflammatory skin symptoms and psychological comorbidities in the patients.

Coptisine is one of the major alkaloids of Coptidis Rhizoma (also known as Huanglian in China), a traditional medicinal herb used to treat various inflammatory diseases, including psoriasis and atopic dermatitis [20,21,22]. Coptisine has anti-inflammatory, anticancer, and antibacterial effects by regulating different signaling pathways [23]. Previous studies indicated the anti-inflammatory effects of coptisine by inhibiting the production of nitric oxide (NO) and TNF-α in lipopolysaccharide (LPS)-stimulated RAW 264.7 macrophages and LPS-treated rats [24,25]. Coptisine also suppressed the IL-1β-induced expression of cyclooxygenase-2 (COX-2), NO, and NO synthase (iNOS) by inhibiting NF-κB activation in human chondrocytes [26]. In addition, coptisine revealed its neuroprotective effects by decreasing apoptosis in human neuroblastoma cells [27], highlighting its potential for treating neuropsychiatric disorders, including anxiety and depression. These data suggest that coptisine might have beneficial effects on psoriasis by alleviating the inflammatory response and reducing the psychological comorbidities of psoriasis. This study examined the benefits of coptisine on skin inflammation and behavioral changes, as well as the underlying mechanisms using psoriasis models.

## 2. Results

### 2.1. Effects of Coptisine on Psoriasis-like Skin Symptoms in IMQ-Treated Mice

Imiquimod (IMQ) application into mouse dorsal skins induced psoriasis-like symptoms, including scaly and thickened skin lesions. Treatment with coptisine alleviated the severity of skin symptoms and decreased the psoriasis area severity index (PASI) scores compared to the IMQ group (Figure 1A,B). The spleen index was also significantly higher in the IMQ -treated mice, but this was reduced by coptisine application (Figure 1C). The hematoxylin and eosin (H&E) staining results showed that topical treatment of coptisine significantly lowered IMQ-induced epidermal hyperplasia (Figure 1D). Moreover, coptisine also decreased the levels of Th1 and Th17 cytokines, including TNF-α, IL-17, and IL-22 in skin lesions, compared to the IMQ-treated mice (Figure 1E).

### 2.2. Effects of Coptisine on the Behavioral Changes in IMQ-Treated Mice

EPM and TST were conducted to examine the effects of coptisine on the behaviors in IMQ-induced mice. As shown in Figure 2A,B, the IMQ application significantly triggered anxiety-like behavior by decreasing the entries, time, and distance traveled in open arms during the EPM test, compared to the normal mice. In contrast, the coptisine treatment recovered the anxiety-like behavior induced by IMQ (Figure 2A,B). As shown in Figure 2C, IMQ stimulation increased the immobility time in TST, but the difference between the normal and IMQ-treated mice was not significant. Coptisine also did not affect the immobility time (Figure 2C).

### 2.3. Correlation between the Anxiety-like Behavior and Psoriasis Severity

Previous studies showed that psoriasis could cause anxiety, and in turn, anxiety might exacerbate psoriasis symptoms [28,29]. The correlation between anxiety-like behavior and psoriasis symptoms was confirmed using Pearson’s correlation. Figure 3 shows that entries in open arms were negatively correlated with the PASI score and the skin levels of IL-17 and TNF-α. Similarly, the time in open arms had negative correlations with the PASI score and skin levels of IL-22 and TNF-α (Figure 3).

### 2.4. Effects of Coptisine on the Serum Corticosterone (CORT) Levels and Neuroinflammation in IMQ-Treated Mice

A previous study reported that activation of the hypothalamus–pituitary–adrenal axis, which leads to the secretion of the stress hormone CORT, is related to psychiatric disorders, including anxiety [30]. In this study, the IMQ-stimulated mice exhibited anxiety-like behavior and had a higher level of serum CORT than the normal mice, but the difference was not significant (Figure 4A). In contrast, treatment with coptisine reduced CORT significantly in IMQ-treated mice. Neuroinflammation also plays a crucial role in the pathophysiology of anxiety [31]. IMQ stimulation increased the levels of proinflammatory cytokines TNF-α and IL-1β significantly in the PFC, which were lowered by the coptisine treatment (Figure 4B).

### 2.5. Effects of Coptisine on the Production of Inflammatory Cytokines and Chemokines in M5-Treated HaCaT Cells

A mixture of five inflammatory molecules (TNF-α, IL-1α, IL-17A, IL-22, and oncostatin M; 10 ng/mL each) was used to stimulate HaCaT cells to mimic the microenvironment of psoriasis, as described previously [32]. Pretreatment with coptisine (25, 50 µM) reduced M5-induced production of IL-6, MIP-3α/CCL20, IP-10/CXCL10, and ICAM-1 significantly with no cytotoxic effects (Figure 5A,B). The effects of coptisine on the NF-κB signaling pathway in M5-treated HaCaT cells were examined to determine the underlying mechanism. Figure 5C shows that preincubation with coptisine suppressed NF-κB activation significantly by decreasing the level of p-IKK and increasing the level of IκB-α in the cytoplasm, as well as inhibiting the nuclear translocation of NF-κB p65.

### 2.6. Effects of Coptisine on the Production of Inflammatory Cytokines in LPS-Stimulated BV2 Cells

The anti-neuroinflammatory effects of coptisine in vitro were investigated by stimulating the BV2 cells with LPS to induce inflammation. LPS in BV2 cells increased the production of proinflammatory cytokines, including TNF-α and IL-1β, which were reduced by the coptisine pretreatment at 25 and 50 μM (Figure 6B). Figure 6A shows that coptisine did not have toxic effects on the viability of BV2 cells.

## 3. Discussion

Psoriasis is a common inflammatory skin condition associated with numerous severe comorbidities, including psychiatric disorders, such as anxiety and depression [33]. The amelioration of both psoriasis symptoms and the comorbidities might be a more effective therapeutic approach for psoriasis. Previous studies reported that natural products exerted significant antipsoriatic effects with fewer side effects than other indications (corticosteroids and phototherapy) [34,35]. Moreover, natural compounds commonly possess many biological properties, indicating their broad application in disease treatments [36]. Hence, natural chemicals might be promising candidates to improve skin inflammation and comorbid conditions in patients with psoriasis.

Coptisine, a natural alkaloid from the rhizome of *Coptis* spp. (also known as Coptidis Rhizoma), which has been traditionally used to treat various diseases owing to its ability to clear heat, remove dampness, and detoxify [20]. In traditional medicine, blood-heat syndrome is a typical symptom of psoriasis, and depression can cause internal heat [37,38]. Coptidis Rhizoma with heat-clearing properties could have beneficial effects on both psoriasis symptoms and associated depression. Therefore, coptisine, as the main component of Coptidis Rhizoma, was used to determine its antipsoriatic and antidepressant effects in a psoriasis mouse model in this study.

IMQ-treated mice are a suitable animal model for psoriasis studies with symptoms resembling those in humans [39]. The present study showed that coptisine alleviated psoriasis-like manifestations, including redness, thickening, and scaling in IMQ-induced mice. Psoriatic skin lesions are represented by the infiltration of various inflammatory cells, including dendritic cells, T cells, neutrophils, and macrophages. These cells secrete proinflammatory cytokines to maintain chronic inflammation and promote epidermal hyperplasia in skin lesions [4]. Infiltrated dendritic cells produce several inflammatory molecules, such as IL-12 and IL-23, which activate the maturation of naïve T cells to Th1, Th17, and other subtypes of T cells [40]. Activated Th1 cells can produce proinflammatory molecules, TNF-α and IFN-γ, and interact with dendritic cells to trigger type 1 inflammation in psoriasis [41]. Th17 cells with their derived cytokines, IL-17 and IL-22, participate in the progression of psoriasis by inducing inflammation and keratinocyte proliferation [42]. In this study, the topical treatment of coptisine reduced the IMQ-induced secretion of Th1/Th17 cytokines, including TNF-α, IFN-γ, IL-17, and IL-22 in skin lesions. Treatment with coptisine also reduced epidermal thickening in the IMQ-stimulated mice. These results agree with a previous study reporting that a coptisine derivative (coptisine free base) also exerted anti-inflammatory effects in edema mouse models [43]. The doses of coptisine used in this study were 0.05% and 0.5%, which were similar to the common corticosteroid doses for the topical treatment of psoriasis [44]. The effects of coptisine were also comparable to clobetasol, a commercially available steroid indicated for psoriasis.

Keratinocytes are the primary cells in the skin that play a crucial role in the progression of psoriasis. In the early phase, keratinocytes can produce a variety of cytokines and chemokines, including IL-6, IP-10/CXCL10, MIP-3α/CCL20, and ICAM-1, to recruit inflammatory cells into skin lesions [45]. IL-6 is upregulated in psoriatic skin and contributes to neutrophil recruitment into the inflamed areas [46,47]. The expressions of IP-10/CXCL10 and MIP-3α/CCL20, which are chemoattractants for Th1 and Th17 cells, also increased in psoriatic keratinocytes [6,48,49]. ICAM-1 is an important adhesion molecule for T-cell recruitment in psoriatic skin lesions, and the overexpression of ICAM-1 was observed in epidermal keratinocytes from psoriasis patients [50]. In the current study, the levels of IL-6, IP-10/CXCL10, MIP-3α/CCL20, and ICAM-1 were upregulated in M5-stimulated HaCaT cells (in vitro model of psoriasis). These increases were reduced by pretreatment with coptisine, highlighting its antipsoriatic potential by suppressing the production of inflammatory mediators.

NF-κB is a key signaling pathway that regulates the inflammatory response in psoriasis [51]. The NF-κB exists in the cytoplasm in an inactive complex bound to IκB-α. Upon cytokine stimuli, IκB kinase (IKK) was phosphorylated at specific sites of IKKα and IKKβ subunits. Phosphorylated IKK could degrade IκB-α and liberate NF-κB from the complex with IκB-α, and triggering the nuclear translocation of NF-κB to regulate various downstream targets, including inflammatory molecules [52]. The high expression of NF-κB was reported in skin lesions from psoriasis patients, and the NF-κB signaling pathway was also highly activated in psoriatic keratinocytes [53,54]. The inhibition of NF-κB signaling is a potential therapeutic approach for psoriasis [55]. In this study, M5 stimulation could induce NF-κB activation in HaCaT cells by increasing the p-IKK level, decreasing the IκB-α level in the cytoplasm, and increasing NF-κB p65 nuclear translocation, which was recovered by pretreatment with coptisine, suggesting that the anti-inflammatory effects of coptisine could be mediated by regulating the NF-κB pathway. These findings concur with a previous study reporting that daphnetin, a natural compound from *Daphne odora*, reduced M5-induced inflammation in HaCaT cells by inhibiting the NF-κB signaling pathway activation [56].

Psoriasis symptoms can induce stress, leading to stress-related conditions, such as depression and anxiety. In turn, the psychological stress might worsen the severity of psoriasis. A cross-sectional study reported anxiety in approximately 17% of psoriasis patients and a positive correlation with the dermatology life quality index (DLQI) [57]. In this study, an EPM test was performed to evaluate anxiety-like behavior in psoriasis mice and the beneficial effects of coptisine. EPM is a valuable behavioral test for screening anxiolytic candidates based on avoidance/approach conflict of mice to open spaces, where avoidance to open arms is considered an anxiety response [58]. The present study showed that IMQ-induced mice decreased in many entries, as well as the time in open arms, demonstrating anxiety-like behavior in these psoriasis mice. These results are consistent with a previous study showing that the K5.Stat3C psoriasis mouse model also showed anxiety behavior in the EPM and open field test [59]. The topical application of coptisine had anxiolytic effects by increasing the open arm entries and time spent in open arms in EPM. Moreover, the severity of anxiety behavior strongly correlated with psoriasis symptoms, including the PASI score and skin inflammation. These findings suggest that the coptisine treatment had beneficial effects on anxiety-like behavior caused by psoriasis symptoms. A previous study reported that mice with IMQ-induced psoriasis showed depression-like behavior in the sucrose preference test (SPT) [60]. This study indicated that the immobility time of IMQ-treated mice in TST was higher than the normal mice, but the difference was not significant, demonstrating that the depression behavior was unclear in psoriasis mice. The inconsistency between the two studies might be due to the different behavioral tests. The aforementioned study conducted SPT, which measured the anhedonia behavior, while this study used TST, which tests the despair behavior in mice [61]. This might lead to different outcome results. Hence, further studies will be needed to confirm that IMQ can induce depression-like behavior in mice.

Neuroinflammation is believed to cause neuropsychiatric disorders and is considered a potential biomarker of anxiety [31]. A previous study reported that anxiety-like behavior in obese mice was associated with increased levels of TNF-α, IL-1β, and IL-6 in the brain [62]. In contrast, treatment with TNF-α and IL-1β inhibitors could suppress the development of anxiety in mouse models [63,64]. In the current study, elevated levels of proinflammatory molecules TNF-α and IL-1β were observed in the prefrontal cortex of IMQ-induced psoriasis mice. On the other hand, the coptisine treatment significantly alleviated the production of these cytokines, suggesting the anxiolytic effects of coptisine might occur by reducing neuroinflammation. These data concur with a previous report that Coptidis Rhizoma, a coptisine-containing herb, also exerted anti-neuroinflammatory effects in stress-exposed mice [65]. Microglia, which are macrophage-like cells in the central nervous system, play an important role in neuroinflammation and the development of psychiatric disorders [66]. Upon stimuli, microglial cells are activated into two opposite states: M1 (proinflammatory) and M2 (anti-inflammatory). The M1 state of microglia is characterized by the elevated secretion of proinflammatory cytokines, including IL-1β, IL-6, and TNF-α, which trigger neuronal damage and contribute to the development of anxiety and depression [67]. LPS-stimulated BV2 microglial cells were used to mimic neuroinflammatory conditions in vitro [68,69]. In the current study, coptisine inhibited the LPS-induced secretion of TNF-α and IL-1β in BV2 microglial cells, suggesting that coptisine might attenuate psoriasis-associated neuroinflammation by suppressing microglial activation.

In this study, coptisine ameliorated psychiatric disorders despite being topically applied to the skin. In general, murine skin is more permeable than human skin, and the skin barrier of mice used in the experiment is impaired [70]. Therefore, the skin penetration of coptisine might be increased, and it was absorbed from the capillaries of the dermal layer, then acted on the brain through the systemic circulation. Previous studies reported that intravenously administered coptisine can easily cross the blood–brain barrier and enter the brain tissue [71]. Another possible explanation for the effects of coptisine on the central nervous systems is through the brain–skin connection, in that it may send a signal to the brain via the ascending pathway of sensory nerves in the skin or modulate the hypothalamic–pituitary–adrenal axis [72]. Previous reports have demonstrated that chronic topical administration of dermatology drugs such as hydroquinone, isotretinoin, and tacrolimus induce behavioral changes by regulating neuroendocrine systems [73]. Further studies are needed to determine the mechanisms of action of coptisine on central nervous systems.

## 4. Limitation

The present study had some limitations. Some other behavioral tests, e.g., forced swim test or open field test, should be recruited to evaluate the effects of coptisine on psoriasis-induced behavioral abnormalities in mice. In the psoriasis model, further studies are needed to determine whether coptisine suppresses the proportion of Th1 and Th17 cells and regulates associated transcription factors. Coptidis alkaloids, which are present in the aqueous extract of Coptidis Rhizoma, have been demonstrated to inhibit the proliferation of Th1/Th17 and regulatory T (Treg) cells in mouse bone marrow cells by suppressing T-bet (the transcription factor of Th1) and Foxp3 (the transcription factor of Treg) [74]. Furthermore, it has been revealed that herbal formulas containing coptisine also affect Th cell-specific transcription factors such as T-bet and RORγt (the transcription factor of Th17) [75]. Additional studies will be needed to confirm the beneficial effects of coptisine on neuronal cells and neurogenesis in the psoriasis model. In this study, the in vitro experiments focused on keratinocytes and microglial cells. However, coptisine showed neuroprotective effects by suppressing apoptosis and increasing the viability of SH-SY5Y cells under oxidative stress conditions [27].

## 5. Materials and Methods

### 5.1. Animals

Six-week male BALB/c mice (18–20 g) from Laboratory Animal Resource Center (Seoul, Korea) were acclimated for 1 week before the experiment. All animal experiments were performed according to the Institutional Animal Care and Use Committee of Dongguk University protocol (Approval no. IACUC-2020-08). All mice were allowed access to standard food and water ad libitum. A total of 25 mice were divided randomly into five groups (*n* = 5 per group): the normal control group (NC group), imiquimod-treated group (IMQ group), IMQ + 0.05% coptisine-treated group (C0.05 group), IMQ + 0.5% coptisine-treated group (C0.5 group), and IMQ + 0.05% clobetasol-treated group (CLO group). IMQ cream (62.5 mg, 5%) (3M Health Care, Loughborough, UK) was applied to shaved back skins once daily for 6 days. Coptisine (CFN99563, Chemfaces, Wuhan, China) and clobetasol (Sigma-Aldrich, St. Louis, MO, USA) were dissolved in acetone for the topical treatment. The skin was treated with coptisine or clobetasol (100 µL/mouse) for 1 h before applying IMQ. The skin severity was evaluated using the Psoriasis Area Severity Index (PASI) with three criteria: redness, thickness, and scaling. The scoring system included the following: 0 (none), 1 (mild), 2 (moderate), 3 (severe), and 4 (very severe). The mice were anesthetized using isoflurane and sacrificed from 9:00 to 12:00 on the last day of experiment. The body weights and spleen weights were recorded. The spleen index (the spleen weight divided by the body weight) of each mouse was determined. The serum, back skin tissues and prefrontal cortex (PFC) samples were collected for further analysis.

### 5.2. Elevated Plus Maze (EPM)

The effects of coptisine on anxiety-like behavior were examined in IMQ-induced psoriasis mice by conducting an EPM test, as described previously [58]. Briefly, mice were placed in the center zone of the maze (two open arms and two closed arms; arm size: 30 cm × 5 cm), and a 5 min session was recorded using a SMART v3.0 video tracking system (Panlab, Barcelona, Spain). The number of entries in open arms, time in open arms, and distance traveled in open arms were assessed.

### 5.3. Tail Suspension Test (TST)

TST was performed to investigate the effects of coptisine on depression-like behavior in IMQ-induced psoriasis mice, as described previously [76]. Briefly, the tail of the mouse was attached to a bar using adhesive tape (length: 15 cm). The distance from the bar to the ground was fixed at 50 cm. The total immobility time for each mouse was assessed in a 6-min session. The immobility behavior was defined as no movement of the forelimbs.

### 5.4. Histological Examination

Skin tissue samples were collected and prepared for histological staining by fixing in 4% paraformaldehyde and embedding in paraffin. The 5 μm skin sections were produced and stained with a hematoxylin and eosin (H/E) solution. Obtained samples were analyzed using an automated microscope (Lionheart FX, Biotek Instruments Inc., Winooski, VT, USA) at 100× magnification.

### 5.5. Cell Culture

HaCaT cells (an immortalized human keratinocyte cell line) and BV2 cells (a murine microglial cell line) were supported by the Korea Institute of Oriental Medicine (KIOM, Daegu, Korea) and maintained in DMEM high glucose (WELGENE Inc., Gyeongsangbuk, Korea) supplemented with 10% fetal bovine serum (Sigma-Aldrich, St. Louis, MO, USA) and 1% antibiotics (Thermo Fisher Scientific, Waltham, MA, USA) at 37 °C in a 5% CO_2_ humidified incubator. The HaCaT cells were pretreated with coptisine (10, 25, and 50 μM) or clobetasol (10 μM) for 1 h before stimulation with M5 (TNF-α, IL-1α, IL-17A, IL-22, and oncostatin M; 10 ng/mL each) for 24 h. The BV2 cells were pretreated with coptisine (10, 25, and 50 μM) or clobetasol (10 μM) for 1 h before being stimulated with lipopolysaccharide (LPS, 1 µg/mL) for 24 h.

### 5.6. Cell Viability

MTT assays were conducted to determine the cytotoxicity of coptisine on HaCaT and BV2 cells. After treating the cells with coptisine (5, 10, 25, and 50 μM) for 24 h, 10 µL of an MTT solution (5 mg/mL, Sigma-Aldrich, St. Louis, MO, USA) was added and incubated for 4 h at 37 °C in the dark. The supernatants were removed and 50 µL of DMSO was added. The optical density was evaluated at 570 nm using a microplate reader (Tecan, Männedorf, Switzerland).

### 5.7. Enzyme-Linked Immunosorbent Assay (ELISA)

The skin homogenates were prepared using a lysis buffer (Tissue Extraction Reagent I, Thermo Fisher Scientific, Waltham, MA, USA) and centrifuged (10,000× *g*, 20 min) for supernatant collection. The levels of the following cytokines in the cultured supernatants from BV2 cells were assessed using commercial ELISA kits following the manufacturer’s instructions: TNF-α, IFN-γ, IL-17, and IL-22 in the skin samples; CORT in the serum samples; IL-1β, IL-6, and TNF-α in the PFC samples; IL-6, MIP-3α, IP-10, and ICAM-1 in the cultured supernatants from the HaCaT cells; TNF-α and IL-1β. The human IL-6, IP-10/CXCL10, and ICAM-1, and mouse TNF-α, IL-1β, IL-6, IFN-γ, IL-17, and IL-22 ELISA kits were obtained from LABISKOMA (Seoul, Korea). The mouse CORT ELISA kit was obtained from Arigo (Hsinchu, Taiwan, China). The human MIP-3α/CCL20 ELISA kit was purchased from R&D Systems (Minneapolis, MN, USA). The optical density at 450–550 nm was determined using a microplate reader (Tecan, Männedorf, Switzerland).

### 5.8. Western Blot Analysis

The nuclear and cytoplasmic protein fractions from HaCaT cells were extracted using NE-PER Extraction Reagents (Thermo Fisher Scientific, Waltham, MA, USA), according to the protocol from the manufacturer. The protein concentration was evaluated using Bradford assays. Subsequently, 20 µg of the total proteins were resolved using 10% SDS-PAGE electrophoresis and transferred to PVDF membranes (Merck Millipore, Carrigtwohill, Ireland). The membranes were blocked in a 5% skim milk solution, then incubated with the primary antibodies, followed by secondary antibody horseradish peroxidase (HRP)-conjugated anti-IgG. The primary antibodies included anti-NF-κB p65 (#8242S, Cell Signaling Technology, Danvers, MA, USA), p-IKK (#2694S, Cell Signaling Technology), IκB-α (#4814S, Cell Signaling Technology), lamin B2 (ab151735, Abcam, Cambridge, UK), and β-actin (A1978, Sigma-Aldrich, St. Louis, MO, USA). The HRP-conjugated goat anti-rabbit and goat anti-mouse IgG secondary antibodies were procured from Sigma-Aldrich (St. Louis, MO, USA). All blots were visualized using ECL reagents and ChemiDoc imaging systems (BioRad, Hercules, CA, USA). The relative intensities of the protein bands were determined using Gel-Pro analyzer software (version 3.1, Media Cybernetics, Rockville, MD, USA).

### 5.9. Statistical Analysis

All experiments were performed in at least three independent experiments. GraphPad Prism v5.03 (GraphPad Software, San Diego, CA, USA) was used for statistical analysis. The results are presented as the means ± standard deviation (SD) followed by the statistical significance (one-way ANOVA with post hoc Tukey’s test or two-tailed unpaired Student’s *t*-test) with a *p*-value < 0.05. Pearson’s correlation analysis was performed to examine the correlation between the two parameters.

## 6. Conclusions

Coptisine alleviated both skin inflammation and associated anxiety-like behavior in the IMQ-induced psoriasis-like mouse model. Moreover, pretreatment with coptisine attenuated the M5-induced proinflammatory cytokine and chemokine production in HaCaT keratinocytes by suppressing the NF-κB signaling pathway. These findings suggest that coptisine might be a potential candidate for psoriasis treatment by improving the disease severity and psychological comorbidities.

## Figures and Tables

**Figure 1 molecules-27-01412-f001:**
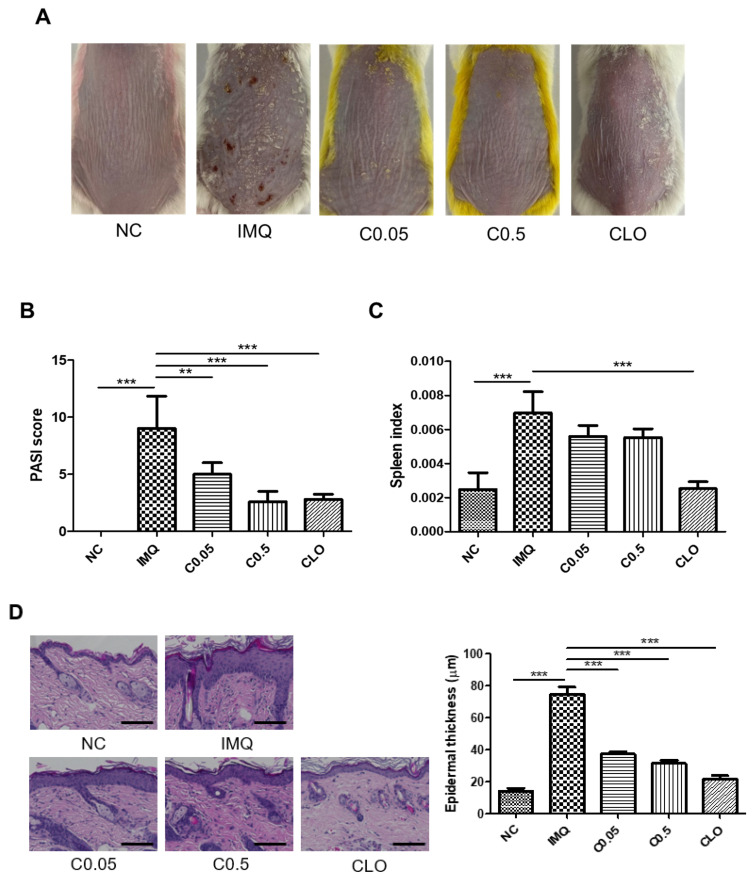
Effects of coptisine on psoriasis-like skin symptoms in IMQ-treated mice. (**A**) Representative skin lesions from mice of five groups. PASI score (**B**) and spleen index (**C**) were measured. (**D**) Representative H&E staining results and quantification of epidermal thickness in skin lesions. Scale bar: 100 μm, magnification: 20×. (**E**) Levels of IL-17, IL-22, TNF-α, and IFN-γ in skin lysates. The data represent the means ± SDs (*n* = 5 per experiment). * *p* < 0.05, ** *p* < 0.01, *** *p* < 0.001 using one-way ANOVA with post hoc Tukey’s test. NC, normal control; IMQ, imiquimod; C, coptisine; CLO, clobetasol.

**Figure 2 molecules-27-01412-f002:**
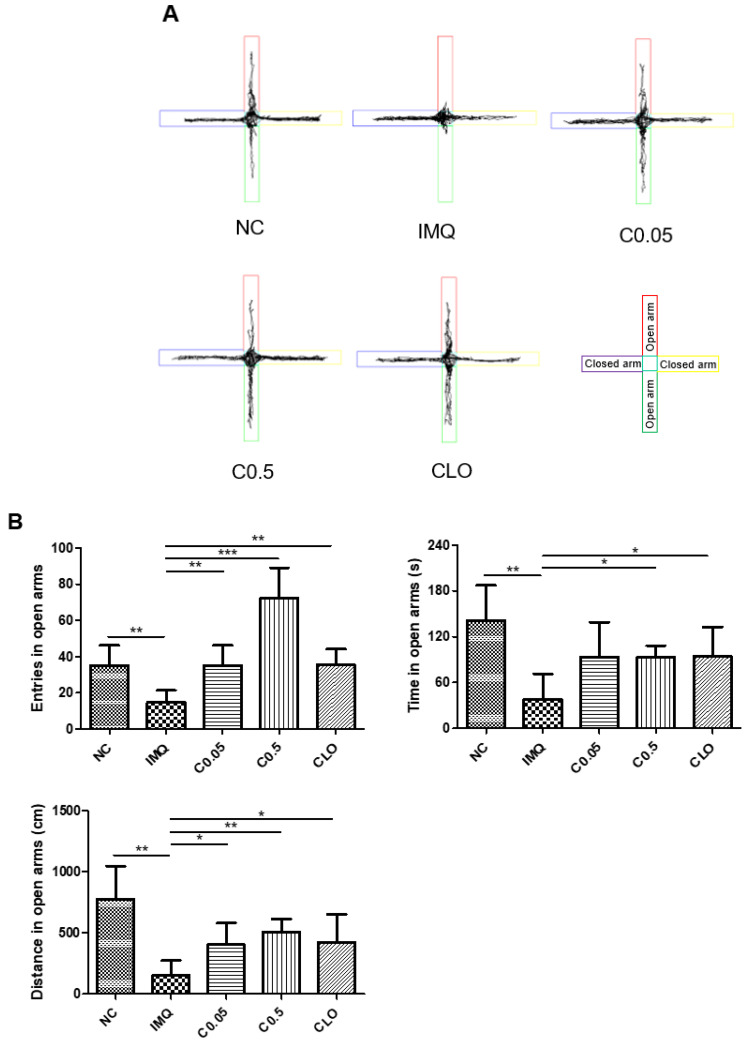
Effects of coptisine on the behavioral changes in IMQ-treated mice. (**A**) Representative trajectory maps in the EPM test. (**B**) The number of entries, time, and distance in open arms in the EPM was recorded. (**C**) The immobility time in TST was recorded. The data represent the means ± SDs (*n* = 5 per experiment). * *p* < 0.05, ** *p* < 0.01, *** *p* < 0.001 using Student’s *t*-test for unpaired experiments. NC, normal control; IMQ, imiquimod; C, coptisine; CLO, clobetasol.

**Figure 3 molecules-27-01412-f003:**
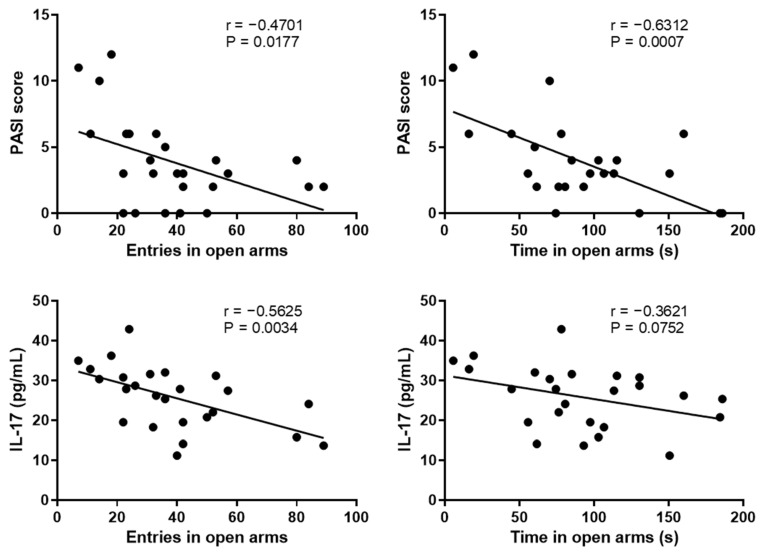
Correlations between the number of entries and time in open arms in the EPM test and PASI score, skin levels of IL-17, IL-22, TNF-α were analyzed using Pearson’s correlation. EPM, elevated plus maze; PASI, psoriasis area severity index.

**Figure 4 molecules-27-01412-f004:**
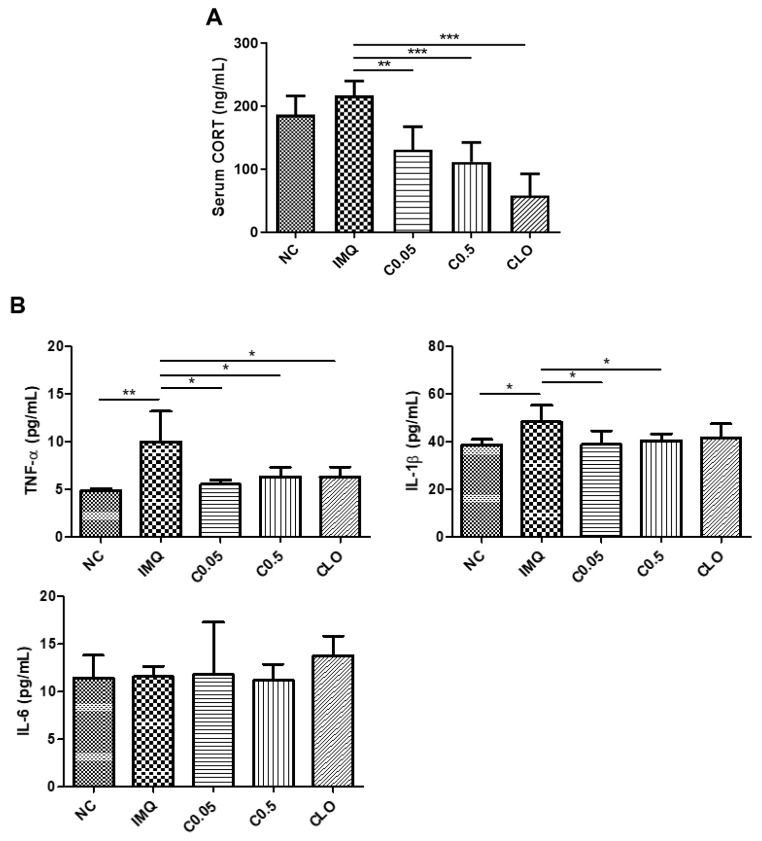
Effects of coptisine on the serum CORT levels and neuroinflammation in IMQ-treated mice. (**A**) Serum levels of CORT. (**B**) Levels of TNF-α, IL-1β, and IL-6 in the PFC. The data represent the means ± SDs (*n* = 5 per experiment). * *p* < 0.05, ** *p* < 0.01, *** *p* < 0.001 using Student’s *t*-test for unpaired experiments. NC, normal control; IMQ, imiquimod; C, coptisine; CLO, clobetasol; CORT, corticosterone.

**Figure 5 molecules-27-01412-f005:**
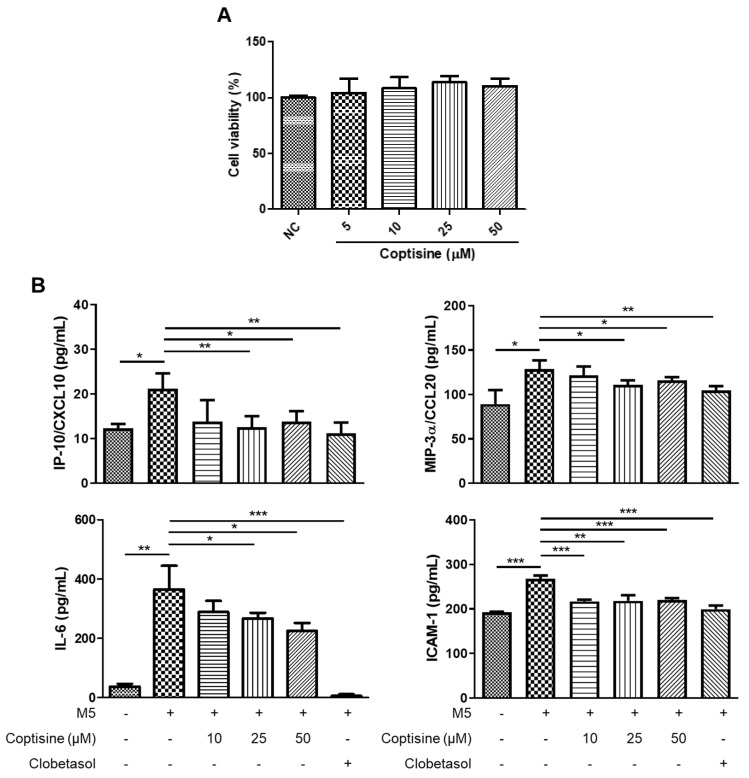
Effect of coptisine on the production of inflammatory cytokines and chemokines in M5-treated HaCaT cells. (**A**) The viability of HaCaT cells was measured using MTT assays. (**B**) The levels of IP-10/CXCL10, IL-6, ICAM-1, and MIP-3α/CCL20 in cell culture media were evaluated using ELISA kits. (**C**) Protein levels of NF-κB, lamin B2 in the nucleus, and protein levels of p-IKK, IκB-α, β-actin in the cytoplasm. The data represent the means ± SDs (*n* = 3 per experiment). * *p* < 0.05, ** *p* < 0.01, *** *p* < 0.001 using Student’s *t*-test for unpaired experiments.

**Figure 6 molecules-27-01412-f006:**
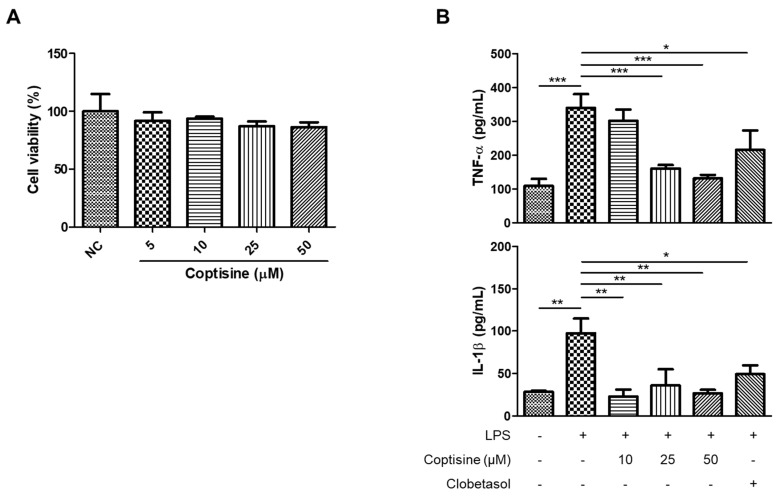
Effect of coptisine on the production of inflammatory cytokines in LPS-stimulated BV2 cells. (**A**) The viability of BV2 cells was measured using MTT assays. (**B**) The levels of TNF-α and IL-1β in cell culture media were evaluated using ELISA kits. The data represent the means ± SDs (*n* = 3 per experiment). * *p* < 0.05, ** *p* < 0.01, *** *p* < 0.001 using Student’s *t*-test for unpaired experiments.

## Data Availability

The data presented in this study are available in this article.

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
