# Peer review of "Coptisine Alleviates Imiquimod-Induced Psoriasis-like Skin Lesions and Anxiety-like Behavior in Mice"

_molecules, 2022, doi:10.3390/molecules27041412_

Round 1
Reviewer 1 Report
The manuscript assesses the efficacy and the mechanism of action of a natural compound coptisine using imiquimod (IMQ)-induced psoriasis mice. After carefully reading the manuscript, I conclude that the abstract, introduction, and other chapters cover the issues discussed in an extensive and proper manner. The conclusions presented by the authors are consistent with the evidence and relate to the main research issue. Although it is a valuable work having an interesting idea it needs some adjustment that I list below.
DETAILED REVIEW
I strongly recommend separating the description of study limitations from the discussion in the form of an additional chapter. Additionally, since a large number of abbreviations are used along the text, I recommend that all used abbreviations should be listed at the end of the main body of text before references. This will certainly provide the reader with a better understanding of the context of the issues discussed. In addition, this is a kind of standard in contemporary scientific papers.
I recommend publication after minor revision.
Author Response
We would like to thank for the constructive and useful comments on the manuscript.
In the following, we give a point-by-point reply to your comments.
REVIEWER 1
Comments: the manuscript assesses the efficacy and the mechanism of action of a natural compound coptisine using imiquimod (IMQ)-induced psoriasis mice. After carefully reading the manuscript, I conclude that the abstract, introduction, and other chapters cover the issues discussed in an extensive and proper manner. The conclusions presented by the authors are consistent with the evidence and relate to the main research issue. Although it is a valuable work having an interesting idea it needs some adjustment that I list below.
Response: Thanks for your constructive and helpful advice.
Comments: I strongly recommend separating the description of study limitations from the discussion in the form of an additional chapter. Additionally, since a large number of abbreviations are used along the text, I recommend that all used abbreviations should be listed at the end of the main body of text before references. This will certainly provide the reader with a better understanding of the context of the issues discussed. In addition, this is a kind of standard in contemporary scientific papers. I recommend publication after minor revision.
Response: Thanks for your constructive and helpful advice. As you suggested, we separated the description of study limitations from the discussion in the form of an additional chapter. Abbreviations were listed at the end of the main body of the text before references.

Reviewer 2 Report
The paper investigates various effects of coptisine, one of the major alkaloids of Coptidis Rhizoma (also known as 49 Huanglian in China) on imiquimode induced inflammation in mice. There is no problem with the experiments and their results, all are done and presented well. I have some concerns, namely it is not true that ”corticosteroids, one of the most common treatments for psoriasis”. Corticosteroids administered systemically is not really used in psoriasis, in fact, it is advised not to use it, with some rare exceptions. It is mostly used topically in patients with mild disease. The likelihood of psychological stress and psychiatric disorders related to corticosteroid therapy has been described in relation with systemic therapy only. There are numerous other treatments that are used systemically to treat psoriasis, those should be mentioned and their effects on inflammation depression and anxiety should be discussed.
This paper maybe of interest to the authors: Vasilakis-Scaramozza C, Persson R, Hagberg KW, Jick S. The risk of treated anxiety and treated depression among patients with psoriasis and psoriatic arthritis treated with apremilast compared to biologics, DMARDs and corticosteroids: a cohort study in the United States MarketScan database. J Eur Acad Dermatol Venereol. 2020 Aug;34(8):1755-1763.
In Figure 1. one-way ANOVA with post hoc Tukey’s test was used, on the other hand in Figure 5 Student’s t-test for unpaired experiments. Why?
There is some confusion regarding the IkB-alfa and NF-kB data both in the results and the discussion sections.
Author Response
We would like to thank for the constructive and useful comments on the manuscript. In the following, we give a point-by-point reply to your comments.
REVIEWER 2
Comments: The paper investigates various effects of coptisine, one of the major alkaloids of Coptidis Rhizoma (also known as 49 Huanglian in China) on imiquimode induced inflammation in mice. There is no problem with the experiments and their results, all are done and presented well. I have some concerns, namely it is not true that “corticosteroids, one of the most common treatments for psoriasis”. Corticosteroids administered systemically is not really used in psoriasis, in fact, it is advised not to use it, with some rare exceptions. It is mostly used topically in patients with mild disease. The likelihood of psychological stress and psychiatric disorders related to corticosteroid therapy has been described in relation with systemic therapy only. There are numerous other treatments that are used systemically to treat psoriasis, those should be mentioned and their effects on inflammation depression and anxiety should be discussed.
This paper maybe of interest to the authors: Vasilakis-Scaramozza C, Persson R, Hagberg KW, Jick S. The risk of treated anxiety and treated depression among patients with psoriasis and psoriatic arthritis treated with apremilast compared to biologics, DMARDs and corticosteroids: a cohort study in the United States MarketScan database. J Eur Acad Dermatol Venereol. 2020 Aug;34(8):1755-1763.
Response: Thanks for your constructive and helpful advice. Following your advice, we have mentioned other treatments and their effect on psoriasis.
Comments: In Figure 1. one-way ANOVA with post hoc Tukey’s test was used, on the other hand in Figure 5 Student’s t-test for unpaired experiments. Why?
Response:
Thanks for your constructive and helpful advice. Both ANOVA and Student’s t-test are effective to observe statistical significance between groups. In Figure 5, we followed a previous report that the Student’s t-test has the highest power compared to other statistical tests for small sample sizes of 3 or 4 per group (J Neurosci Methods. May 15, 2009; 179(2):173-8).
Comments: There is some confusion regarding the IkB-alfa and NF-kB data both in the results and the discussion sections.
Response: Thanks for your constructive and helpful advice. We changed the descriptions of the IkB-alfa and NF-kB data in the results and discussion section.

Reviewer 3 Report
This is an interesting study focusing on depression/anxiety associated with psoriasis. Several issues should be solved for its publication
1.How coptisine inhibits NF-kB should be detailed. Does the medicine suppress IKK? In Fig.5, the levels of phosphorylated IKK or phosphorylated IkBa should be shown.
2.Which cell types are the target of coptisine? Other than keratinocytes and microglia cells, does coptisine directly act on T cells, dendritic cells, or neurons, directly suppress IL-17/IL-22 secretion in T cells? This should be discussed.
3.In the series of experiments, coptisine is applied topically to the imiquimod-applied skin. Does topically applied coptisine systemically circulate to central nervous systems? This should be discussed.
4.Fig. 3: Pearson's correlation efficient may be shown as R. However, R2 is shown in the figure. Is that type error?
5.Fig.4: serum cortisol level may change dependently the time of whole day. That in the morning may be higher than in the evening. When was the cortisol level measured?
6.The authors only referred to NF-kB as the target molecule of coptisine. What about other transcription factors such as RORgt, T-bet, Foxp3? Those should be discussed.
Author Response
We would like to thank for the constructive and useful comments on the manuscript.
In the following, we give a point-by-point reply to your comments.
REVIEWER 3
Comments: This is an interesting study focusing on depression/anxiety associated with psoriasis. Several issues should be solved for its publication
Response: Thanks for your constructive and helpful advice. In the following, we give a point-by-point reply to your comments.
Comments: How coptisine inhibits NF-kB should be detailed. Does the medicine suppress IKK? In Fig.5, the levels of phosphorylated IKK or phosphorylated IkBa should be shown.
Response: Thank you for your helpful suggestions. The additional experiment you recommended was carried out, and the results are presented in fig 5.
Comments: Which cell types are the target of coptisine? Other than keratinocytes and microglia cells, does coptisine directly act on T cells, dendritic cells, or neurons, directly suppress IL-17/IL-22 secretion in T cells? This should be discussed.
Response: The study's limitations, as well as additional potential coptisine therapeutic targets, have been included in the manuscript.
Comments: In the series of experiments, coptisine is applied topically to the imiquimod-applied skin. Does topically applied coptisine systemically circulate to central nervous systems? This should be discussed.
Response: Thank you for your helpful suggestions. Following your advice, we have discussed the mechanism of action of topically applied coptisine.
Comments: Fig. 3: Pearson's correlation efficient may be shown as R. However, R2 is shown in the figure. Is that type error?
Response: Thank you for your wise counsel. As you pointed out, we fixed the type error.
Comments: Fig.4: serum cortisol level may change dependently the time of whole day. That in the morning may be higher than in the evening. When was the cortisol level measured?
Response: Thank you for your practical and informative suggestions. Following your advice, we've added serum collection time to the manuscript.
Comments: The authors only referred to NF-kB as the target molecule of coptisine. What about other transcription factors such as RORgt, T-bet, Foxp3? Those should be discussed.
Response: Thank you for your valuable advice. A mention of other transcription factors has been added to the manuscript.
